# Yap1 promotes sprouting and proliferation of lymphatic progenitors downstream of Vegfc in the zebrafish trunk

Lin Grimm[1], Hiroyuki Nakajima[2], Smrita Chaudhury[1], Neil I Bower[1], Kazuhide S Okuda[1], Andrew G Cox[3,4], Natasha L Harvey[5], Katarzyna Koltowska[1,6], Naoki Mochizuki[2], Benjamin M Hogan[1]*

[1]Division of Genomics of Development and Disease, Institute for Molecular Bioscience, The University of Queensland, Brisbane, Australia; [2]Department of Cell Biology, National Cerebral and Cardiovascular Centre Research Institute, Osaka, Japan; [3]Cancer Metabolism Program, Organogenesis and Cancer Program, Peter MacCallum Cancer Centre, Melbourne, Australia; [4]Department of Biochemistry and Molecular Biology, University of Melbourne, Parkville, Australia; [5]Centre for Cancer Biology, University of South Australia, SA Pathology, Adelaide, Australia; [6]Department of Immunology, Genetics and Pathology, Uppsala University, Uppsala, Sweden

*For correspondence:
b.hogan@imb.uq.edu.au

Competing interests: The authors declare that no competing interests exist.

**Abstract** Lymphatic vascular development involves specification of lymphatic endothelial progenitors that subsequently undergo sprouting, proliferation and tissue growth to form a complex second vasculature. The Hippo pathway and effectors Yap and Taz control organ growth and regulate morphogenesis and cellular proliferation. Yap and Taz control angiogenesis but a role in lymphangiogenesis remains to be fully elucidated. Here we show that YAP displays dynamic changes in lymphatic progenitors and Yap1 is essential for lymphatic vascular development in zebrafish. Maternal and Zygotic (MZ) *yap1* mutants show normal specification of lymphatic progenitors, abnormal cellular sprouting and reduced numbers of lymphatic progenitors emerging from the cardinal vein during lymphangiogenesis. Furthermore, Yap1 is indispensable for Vegfc-induced proliferation in a transgenic model of Vegfc overexpression. Paracrine Vegfc-signalling ultimately increases nuclear YAP in lymphatic progenitors to control lymphatic development. We thus identify a role for Yap in lymphangiogenesis, acting downstream of Vegfc to promote expansion of this vascular lineage.
DOI: https://doi.org/10.7554/eLife.42881.001

## Introduction

During embryonic development, the lymphatic vascular network derives chiefly from pre-existing veins. In the E9-9.5 mouse embryo, complex, multicellular lymphatic vessels are ultimately generated from a limited pool of lymphatic endothelial cell (LEC) progenitors first specified along the cardinal veins (*Oliver and Srinivasan, 2010*; *Wigle and Oliver, 1999*). LEC progenitors sprout and progressively colonise embryonic tissues over developmental time (*Hägerling et al., 2013*; *Koltowska et al., 2013*). In the zebrafish trunk, a complex lymphatic vasculature is established within just a few days and from a very limited number of progenitors (*Koltowska et al., 2015a*; *Nicenboim et al., 2015*; *Shin et al., 2016*). Beginning from around 32–34 hr post fertilisation (hpf),

lymphangiogenesis generates a functional lymphatic network by 5 days post fertilisation (dpf) (*Hogan and Schulte-Merker, 2017*). To allow this rapid development, multiple molecular pathways have to instruct and orchestrate cellular behaviour as well as control the growth and proliferation of the developing tissue.

Precursor cells in the cardinal vein obtain lymphatic identity upon induction of Prox1 expression, a transcription factor that is essential for development of LECs in both mice and zebrafish (*Johnson et al., 2008*; *Koltowska et al., 2015a*; *Wigle and Oliver, 1999*). Sprouting of LECs from the cardinal veins is governed by Vegfc signalling acting through the endothelial receptor Vegfr3 (Flt4) in both mice and zebrafish (*Hogan et al., 2009b*; *Karkkainen et al., 2004*; *Le Guen et al., 2014*; *Veikkola et al., 2001*; *Villefranc et al., 2013*). Zebrafish Vegfc and Vegfd also play partially compensatory roles in both trunk and craniofacial lymphangiogenesis (*Astin et al., 2014*; *Bower et al., 2017*). Underlining the conservation of this pathway in lymphangiogenesis, mutations in *VEGFC* and *VEGFR3* cause primary lymphedema in humans (*Gordon et al., 2013*; *Irrthum et al., 2000*; *Karkkainen et al., 2000*). Ultimately, induction of Vegfr3 signalling in LECs by Vegfc triggers the activation of multiple downstream intracellular signalling events involved in cell migration, survival and cellular proliferation (*Deng et al., 2015*; *Zheng et al., 2014*). In the zebrafish, Vegfc-Flt4 signalling acts to induce Prox1 expression at the earliest stages of lymphatic specification (*Koltowska et al., 2015a*; *Shin et al., 2016*), although a role in specification remains to be fully explored in the mouse model. Precisely how LECs contextually interpret growth factor signals and elicit a number of different, specific cellular responses still remains to be fully understood.

Key regulators of normal and pathological organ and tissue growth are the Hippo pathway and effector transcription factors, YAP and TAZ, which have been shown to promote proliferation, suppress apoptosis and modulate cellular and tissue morphogenesis (*Harvey et al., 2013*). YAP and its paralogue TAZ are transcriptional co-factors that drive target gene expression by binding to the TEAD1-4 transcriptional co-factors (*Pobbati and Hong, 2013*; *Yu and Guan, 2013*; *Yu et al., 2015*). As potent drivers of cell proliferation, YAP and TAZ have been implicated as oncogenes that are commonly upregulated in various cancer types including colon and breast cancer (*Harvey et al., 2013*; *Varelas, 2014*). Thus, their activity has to be tightly regulated. The classical HIPPO pathway inhibits YAP/TAZ signalling by retaining the effectors in the cytoplasm through the activation of a phosphorylation cascade. The kinase MST1/2 phosphorylates another kinase LATS1/2, which subsequently phosphorylates YAP/TAZ. This phosphorylation sequesters YAP/TAZ in the cytoplasm and leads to their degradation (*Yu and Guan, 2013*; *Yu et al., 2015*). Ultimately, YAP and TAZ function in the cytoplasm, at cell-cell junctions and in the nucleus as core integrators of extracellular stimuli such as growth factor signalling, mechanical forces and cellular adhesion (*Panciera et al., 2017*; *Varelas, 2014*).

Recent studies have demonstrated that YAP and TAZ play important roles in vasculature (*Kim et al., 2017*; *Neto et al., 2018*; *Wang et al., 2017*). While *Yap* knockout mice are lethal due to developmental arrest of the embryo and severe defects in the yolk sac vasculature (*Morin-Kensicki et al., 2006*), endothelial specific deletion of *Yap* and *Taz* leads to vascular defects due to impaired EC sprouting and proliferation (*Kim et al., 2017*; *Neto et al., 2018*). In endothelial cells (ECs), nuclear YAP/TAZ promotes proliferation and cell survival while retention of YAP/TAZ in the cytoplasm leads to apoptosis (*Panciera et al., 2017*; *Zhao et al., 2011*). Moreover, it has been suggested that YAP/TAZ in blood vascular ECs regulate angiogenesis downstream of VEGFA both by modulating cellular proliferation and controlling adherens junctional dynamics during vessel morphogenesis (*Neto et al., 2018*; *Wang et al., 2017*). Roles for Yap and Taz have been recently shown in lymphatic vessel morphogenesis in development and postnatal settings in mice, but the mechanisms of action remain to be fully appreciated (*Cho et al., 2019*). YAP has further been found to respond to altered flow patterns in zebrafish and in cultured blood and lymphatic ECs (*Nakajima et al., 2017*; *Sabine et al., 2015*). Yap1 also contributes to blood vessel maintenance in zebrafish, although blood vessels still undergo normal angiogenesis in zebrafish *yap1* mutant models (*Nakajima et al., 2017*). Despite important roles in the vasculature, in the context of early embryonic lymphatic vascular development, roles for Yap and Taz remain to be fully explored.

Here we utilise zebrafish mutants and live imaging of zebrafish reporters of YAP activity to show that Yap1 is indispensable for lymphatic vascular development. Yap1 acts in a cell autonomous manner and is necessary at stages of lymphangiogenesis driven by Vegfc/Flt4 signalling. However, unlike mutants in the Vegfc/Flt4 pathway Yap1 mutants display normal specification of Prox1-positive

lymphatic progenitors coincident with aberrant cellular behaviours during LEC sprouting. We identify a central role for Yap1 in Vegfc-induced EC proliferation and show that the nuclear concentration of YAP changes dynamically in lymphatic progenitors, driven by Vegfc. This work suggests a dynamic signalling mechanism integrating Vegfc/Flt4 signalling with Yap1 control of LEC proliferation and cellular behaviour during trunk lymphatic vascular development.

## Results

### YAP is active in developing lymphatics of the zebrafish trunk

To investigate whether zebrafish LEC progenitors display Yap1/Taz signalling, we made use of the previously published transgenic line, *Tg(fli1:Gal4db-TEAD2DN-2A-mCherry); Tg(UAS:GFP)*, hereafter called the TEAD reporter (*Nakajima et al., 2017*). This line reports expression when active, endogenous Yap1 binds to the synthetic TEAD-Gal4db protein and subsequently drives EGFP expression from a UAS element. Expression was detected throughout the dorsal aorta (DA), the intersegmental vessels (ISVs) and PCV at 30 hpf, parachordal LEC progenitors (PLs, 2 dpf) and the thoracic duct at 5 dpf (*Figure 1A,B*, *Video 1*).

TEAD-reporter activity is observed via a stable EGFP protein, thus does not accurately report dynamic activity of Yap1. Hence, we used *Tg(fli1a:EGFP-YAP);(fli1a:H2B-mCherry)* (*Nakajima et al., 2017*), hereafter referred to as the YAP reporter line, which expresses human YAP fused to EGFP in zebrafish vasculature. YAP is regulated at the level of entry into the nucleus and EGFP-YAP used here has been shown to accurately reflect endogenous YAP (*Bao et al., 2011*; *Nakajima et al., 2017*). We analysed EGFP-YAP fluorescence intensity (average pixel intensity per nucleus) in the nuclei of PLs relative to the intensity of nuclear H2B-mCherry in the same cells using two different methods (*Figure 1C–G*, see Materials and methods for details). Firstly, the EGFP/mCherry fluorescence intensity ratio was calculated in 3D, using automated surface masking of the nucleus and measuring the intensities for both channels spanning the full z-stack (*Figure 1F and G*). Secondly, fluorescence intensity was extracted from a single Z-plane (2D) at the centre of each nucleus (*Figure 1—figure supplement 1A and D*). Both approaches revealed concordant results (*Figure 1—figure supplement 1E*). We found nuclear EGFP-YAP in PL nuclei that was highly variable between PLs within the same embryo (relative to H2B-mCherry), despite both proteins being driven from the *fli1a* promoter (*Figure 1C–G*). Furthermore, we estimated the nuclear to cytoplasmic ratios for EGFP-YAP between PLs (for the same embryos quantified in *Figure 1F*). We found that differences in the nuclear concentration of EGFP-YAP between PLs (calculated as nuclear EGFP-YAP/mCherry) correlated with differences in the nuclear-cytoplasmic ratio of EGFP-YAP (*Figure 1H* and *Figure 1—figure supplement 1B–C*). This suggests that measurement of nuclear concentration can reflect and serve as a proxy for cytoplasm-nuclear changes. Given the above findings, the variable nature of EC cytoplasmic distribution, and relative accuracy of nuclear measurements, all further analyses use changes in nuclear concentration as a proxy for changes in YAP activity.

Interestingly, when we measured nuclear EGFP-YAP intensities using high speed spinning disc microscopy, we found that EGFP-YAP did not change or fluctuate rapidly over short developmental time periods (imaged over 90 min of development) (*Figure 1—figure supplement 1F–F"*, *Video 2*). However, quantification of nuclear EGFP-YAP levels in individual PLs over longer periods of development revealed dynamic changes (*Video 3*, *Figure 1I*, *Figure 1—figure supplement 1G*). Detailed cell tracking analyses showed that these changes were not associated simply with breakdown of the nuclear lamina during cell division (*Figure 1I* (arrow in cell track two indicates a cell division), *Figure 1—figure supplement 1G*). Taken together, these observations suggest that YAP is dynamically regulated in PLs during lymphatic development.

### Yap1 cell autonomously regulates lymphangiogenesis in the zebrafish trunk

We next examined the formation of the lymphatic vasculature in *yap1* mutants. We analysed two different mutant strains: *yap1*[ncv101-/-] mutants have a 25 bp deletion in exon1 leading to a frame shift and stop codon after 71 bp, prior to the TEAD binding domain (*Nakajima et al., 2017*) and *yap1*[mw48-/-] mutants display a 4 bp deletion after 158 bp, truncating the TEAD binding domain and leading to a premature stop codon (*Miesfeld et al., 2015*). Analysis of the two alleles revealed

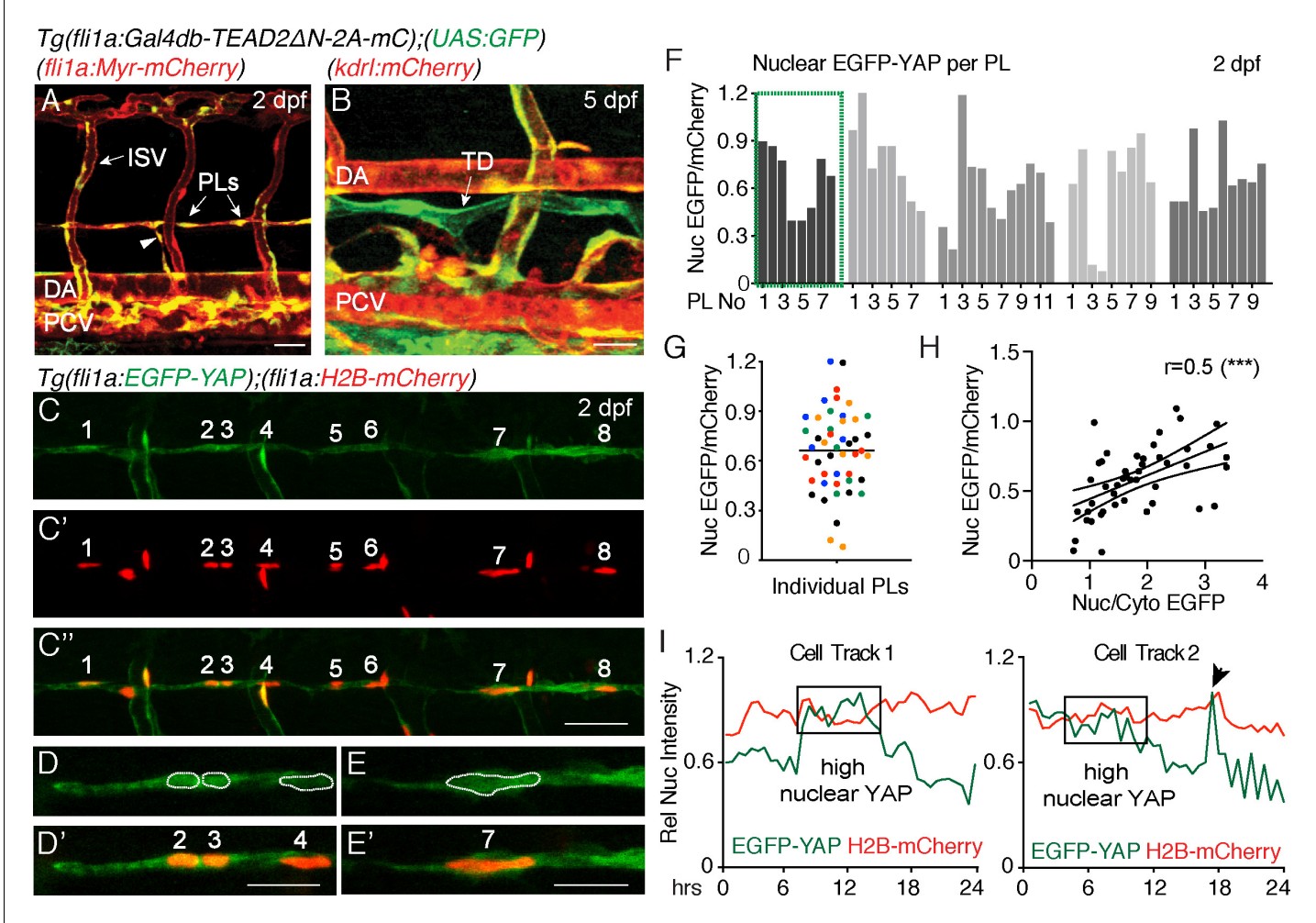

**Figure 1.** Nuclear EGFP-YAP changes dynamically in the developing trunk lymphatic vasculature. (**A–B**) The TEAD reporter line [*Tg(fli1:Gal4db-TEAD2ΔN-2A-mC);(UAS:GFP)*] shows Yap1 activity in vasculature, parachordal LECs (PLs) and cardinal vein sprouts (arrowhead) of the 2 dpf trunk (**A**) as well as in the thoracic duct (TD) at 5 dpf (**B**). Scale bars: 40 µm. (**C**) Maximum projection of 8 PLs in a two dpf embryo showing EGFP-YAP in green [*Tg (fli1:EGFP-YAP)*] and the nucleus in red (**C'**), [*Tg(flil:H2B-mCherry)*] and merge (**C''**). Scale bar: 25 µm. (**D–E**) High power single z-sections of selected PLs from C, showing nuclear YAP in PL2 and PL3, but low nuclear YAP in PL4 (**D–D'**) and PL7 (**E–E'**). Scale bars: 10 µm. (**F**) Quantification of nuclear EGFP/mCherry average pixel intensity across individual PLs from multiple embryos at 2 dpf. Each bar represents a single PL (n = 47), each grey shade a different embryo (n = 5). PLs in (**C** and **D**) highlighted in the green box. EGFP/mCherry Ratios have been calculated using mean fluorescent intensities in 3D. (**G**) Scatter Plot of the Nuclear EGFP/mCherry average pixel intensity for individual PLs (n = 5 embryos). Each colour indicates PLs from a different embryo. Values calculated in 3D measurements of the mean fluorescent intensity for EGFP and mCherry (0.66 ± 0.04, n = 47). (**H**) Pearson Correlation Plot of the Nuclear EGFP/mCherry Ratio values in F and the Nuclear/Cytoplasmic EGFP Ratio values in *Figure 1—figure supplement 1B* (r = 0.52, 95% confidence intervals: 0.27 to 0.70, R square = 0.27, p=0.0002(***)). The two distinct approaches produce correlative measurements. (**I**) Average nuclear pixel intensity graphs from cell tracks of single PLs time-lapse imaged from 2 to 3 dpf. EGFP-YAP intensity (green) is compared over time with H2B-mCherry (red) intensity in individual nuclei. Arrow points to a cell division during time-lapse.

DOI: https://doi.org/10.7554/eLife.42881.002

The following source data and figure supplements are available for figure 1:

**Source data 1.** Measurements of EGFP-YAP in PLs.
DOI: https://doi.org/10.7554/eLife.42881.005
**Figure supplement 1.** Measurements of EGFP-YAP intensity in lymphatic progenitor nuclei over time.
DOI: https://doi.org/10.7554/eLife.42881.003
**Figure supplement 1—source data 1.** Methodological Analysis of EGFP-YAP Intensities in PLs.
DOI: https://doi.org/10.7554/eLife.42881.004

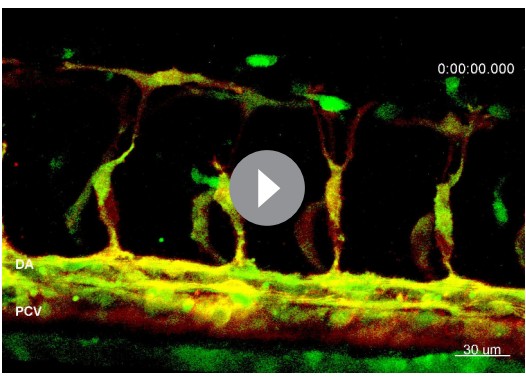

**Video 1.** Time-lapse Video of the developing trunk vasculature of *Tg(fli1:Gal4db-TEAD2ΔN-2A-mC);(UAS: GFP)*] from 30 hpf – 2 dpf. The EGFP signal can be seen to diminish in major trunk blood vessels but remains strong in the dorsal posterior cardinal vein (PCV) and parachordal LECs (PLs).
DOI: https://doi.org/10.7554/eLife.42881.006

equivalent phenotypes. Yap1 is maternally deposited and zygotic *yap1* mutants (Z*yap1^{-/-}*) are viable (*Cox et al., 2016*). Using transgenic lines to visualise developing veins and lymphatics (*TgBAC(dab2b:EGFP)^{ncv67}*, *Tg(kdrl:mCherry)^{ncv502}*), we observed only subtle lymphatic defects in Z*yap1^{ncv101-/-}* mutants with a minor reduction in thoracic duct formation (*Figure 2—figure supplement 1A,C*). To completely deplete the embryo of Yap1, we generated maternal zygotic (MZ*yap1^{-/-}*) mutants in several transgenic backgrounds by crossing heterozygous males to homozygous mutant females. Strikingly, while the overall morphology of MZ*yap1^{-/-}* mutants at 5 dpf appeared relatively normal with mutant embryos displaying subtle craniofacial defects and a missing swim bladder (*Figure 2A*), MZ*yap1^{-/-}* mutants failed to form a mature trunk lymphatic vasculature (*Figure 2B–E*). However, the craniofacial lymphatic network had developed normally (*Figure 2—figure supplement 1B, D*) and blood flow remained intact and functional with apparently normal morphology of blood vessels in the 5 dpf trunk (*Figure 2B–C*). To further investigate whether the MZ*yap1^{-/-}* mutants displayed any abnormalities in the formation of the blood vasculature, we analysed angiogenic sprouting at 24 and 32 hpf (*Figure 2—figure supplement 2*). Compared with wildtype siblings, MZ*yap1^{-/-}* mutants had a transient, mild reduction in the number of EC nuclei in intersegmental vessel (ISV) sprouts (*Figure 2—figure supplement 2A,D,E*), prominent in the most posterior sprouts analysed (*Figure 2—figure supplement 2F*) at 24 hpf. However, the number of ECs in each vessel had largely recovered by 32 hpf (*Figure 2—figure supplement 2B,D,E,F*). Furthermore, we could see normal lumenisation of blood vessels in MZ*yap1^{-/-}* mutants (*Figure 2—figure supplement 2C*).

The above observations indicate that *yap1* is necessary to form a lymphatic vasculature, but not blood vasculature. To test whether Yap1 function is cell autonomous in ECs, we next transplanted wildtype or mutant *Tg(fli1a:EGFP)*

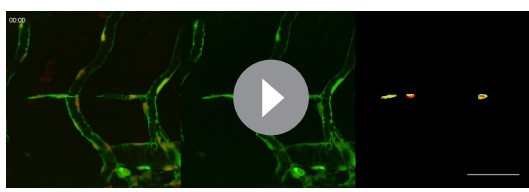

**Video 2.** High-speed time-lapse Video of the developing parachordal LECs (PLs) in [*Tg(fli1a:EGFP-YAP),(fli1a:H2B-mCherry)*] embryos from 2 dpf. EGFP-YAP signal remains relatively stable over the imaged time period of 90 min. Z-stacks were acquired every minute. Fluorescence intensity of EGFP diminishes due to mild bleaching over time. The right panel shows the EGFP fluorescent intensity as a heatmap within the PL nuclei.
DOI: https://doi.org/10.7554/eLife.42881.007

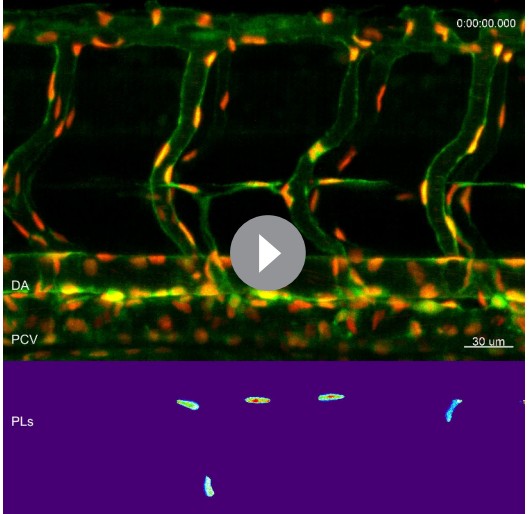

**Video 3.** Time-lapse Video of developing parachordal LECs (PLs) in [*Tg(fli1a:EGFP-YAP),(fli1a:H2B-mCherry)*] embryos from 2 to 3 dpf with images acquired in 20–22 min intervals. The EGFP-YAP signal in individual PLs changes over time as shown in the lower panel displaying a heatmap of the nuclei. Dorsal aorta (DA), posterior cardinal vein (PCV).
DOI: https://doi.org/10.7554/eLife.42881.008

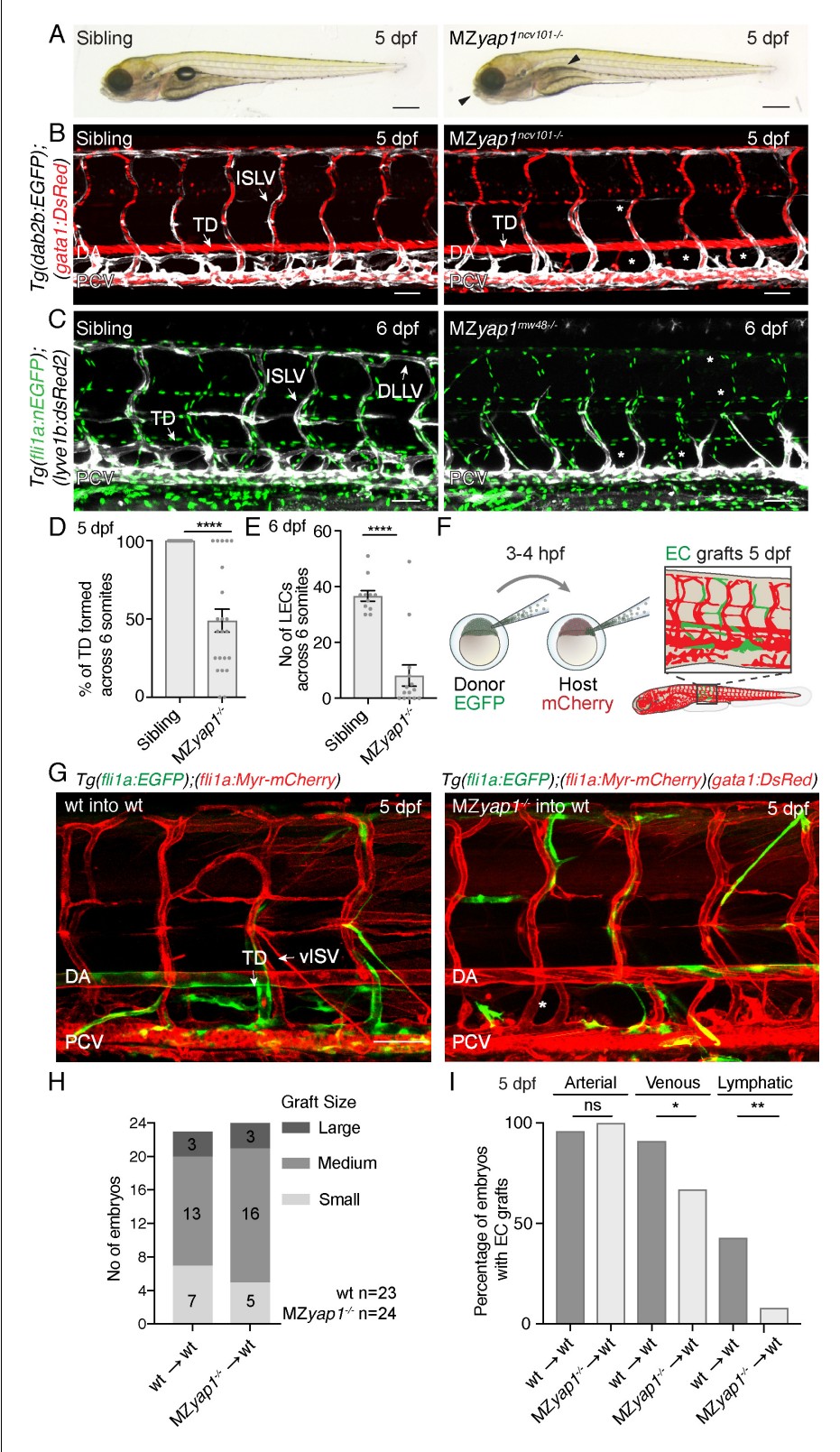

**Figure 2.** Yap1 acts cell autonomously to control trunk lymphangiogenesis in zebrafish. (**A**) Overall morphology of sibling (left) and MZ*yap1*<sup>ncv101-/-</sup> mutant (right) at 5 dpf. Arrowheads indicate mild craniofacial defects and absent swim bladder. Scale bar 200 μm. (**B–C**) Trunk vasculature of sibling and MZ*yap1*<sup>ncv101-/-</sup> mutant embryos at 5 dpf. Veins and lymphatics are displayed in white [*Tg(dab2b:EGFP)*], erythrocytes show normal blood flow in red [*Tg (gata1:DsRed)*]. Asterisks mark absent lymphatic vessels. (**C**) Trunk vasculature of sibling and MZ*yap1*<sup>mw48-/-</sup> mutant at 6 dpf. Vascular nuclei are marked

*Figure 2 continued on next page*

*Figure 2 continued*

in green [*Tg(fli1a:nEGFP)*]. venous and lymphatic vessels in white [*Tg(−5.2lyve1b:DsRed)*]. Asterisks indicate absent lymphatic vessels. Scale bars: 50 μm in B and C. (D) Percentage of TD fragments formed per somite, scored across six somites in total for siblings and MZ*yap1$^{ncv101-/-}$* mutants at 5 dpf (sibling: 100% ± 0, n = 23; MZ*yap1$^{ncv101-/-}$*: 49% ± 7.33, n = 22; p,0.0001). (E) Quantification of the total number of LECs across 6 somites at 6 dpf (sibling: 37 ± 1.91, n = 11; MZ*yap1$^{mw48-/-}$*: 8 ± 3.79; p<0.0001). (F) Schematic showing the cell transplantation technique. Blastomere cells are transplanted from donor (EGFP) into host (mCherry) embryos. This results in a chimeric host embryo (right) with randomly located, transplanted EC grafts. (G) Representative images of host chimeric trunk vessels at 5 dpf. Wildtype (wt) donor ECs contribute to all vascular EC types, while MZ*yap1$^{ncv101-/-}$* mutant ECs show reduced propensity to contribute to lymphatic structures. Asterisk marks missing TD. Scale bars: 50 μm. (H) Graft sizes analysed for vascular grafts. Numbers within bars represent the number of embryos scored for each graft size. (I) Percentage of embryos with EC grafts contributing to arterial (wt into wt: 96% ± 4; MZ*yap1$^{ncv101-/-}$* into wt: 100% ± 0; p=0.31, not significant (ns)), venous (wt into wt: 91% ± 6; MZ*yap1$^{ncv101-/-}$* into wt: 67% ± 10; p=0.04 (*)) and lymphatic vessels (wt into wt: 43% ± 10; MZ*yap1$^{ncv101-/-}$* into wt: 8% ± 6; p=0.005 (**)). Wildtype into wildtype total number of EC grafts: n = 23; MZ*yap1$^{ncv101-/-}$* mutant into wildtype total number of EC grafts: n = 24. Dorsal aorta (DA); Posterior Cardinal Vein (PCV); Thoracic Duct (TD); Intersegmental Lymphatic vessel (ISLV), Dorsal Longitudinal Lymphatic vessel (DLLV).
DOI: https://doi.org/10.7554/eLife.42881.009

The following source data and figure supplements are available for figure 2:

**Source data 1.** Measurements for the lymphatic phenotype of the MZ*yap1$^{-/-}$* mutant.

DOI: https://doi.org/10.7554/eLife.42881.014

**Figure supplement 1.** Z*yap1$^{-/-}$* mutants only exhibit mild lymphatic defects in the trunk and MZ*yap1$^{-/-}$* mutants form facial lymphatics.

DOI: https://doi.org/10.7554/eLife.42881.010

**Figure supplement 1—source data 1.** Quantification of the lymphatic phenotype of the Z*yap$^{-/-}$* mutant trunk and MZ*yap1$^{-/-}$* mutant craniofacial phenotype.

DOI: https://doi.org/10.7554/eLife.42881.011

**Figure supplement 2.** MZ*yap1$^{-/-}$* mutants do not show major defects in blood vessel formation.

DOI: https://doi.org/10.7554/eLife.42881.012

**Figure supplement 2—source data 1.** Characterisation of angiogenic sprouting in the MZ*yap1$^{-/-}$* mutants.

DOI: https://doi.org/10.7554/eLife.42881.013

ECs into wildtype or mutant *Tg(fli1a:myrmCherry)* hosts (*Figure 2F–G*). Subsequently, we selected embryos with EC grafts and scored grafts of similar size (*Figure 2H*). We scored the contribution of ECs in vascular grafted embryos to arterial, venous and lymphatic vessels (*Figure 2I*). Across n = 23 wildtype EC grafts, ECs contributed to arterial and venous structures as well as to lymphatic vessels at frequencies equivalent to those we have previously reported (*Koltowska et al., 2015b*). Across n = 24 mutant EC grafts, MZ*yap1$^{-/-}$* mutant ECs were able to form arterial grafts at frequencies comparable to wildtype EC grafts. However, MZ*yap1$^{-/-}$* mutant ECs formed venous grafts at a slightly reduced frequency and failed to contribute to lymphatic vessels except for small numbers of cells in just n = 2/24 transplanted embryos (compared with n = 10/23 robust contributions in embryos transplanted with WT cells) (*Figure 2G–I*). Overall, this indicates that Yap1 has a cell autonomous role in trunk lymphangiogenesis.

## Yap1 is dispensable for specification but essential for sprouting and Vegfc-induced proliferation of LECs

LEC progenitors sprout from the PCV in a Vegfc/Flt4 dependent manner and colonise the HM transiently, a period when they are highly proliferative as well as migratory (*Bussmann et al., 2010*; *Cha et al., 2012*; *Koltowska et al., 2015a*; *Yaniv et al., 2006*). We scored MZ*yap1$^{-/-}$* mutants for ECs sprouting from the PCV at 2 dpf and found a reduction in the total number of ECs departing the vein (*Figure 3A–B*). Strikingly, we saw almost a complete loss of PLs, but near normal numbers of ECs in venous intersegmental blood vessels (*Figure 3A,C,D*). By 3 dpf, PL numbers in the HM remained reduced but showed recovery compared with sibling controls (*Figure 3E*). To investigate whether this reduction in PL numbers is a result of reduced LEC progenitor specification, we examined Prox1 expression as a marker of LEC fate. We found that Prox1 expression in the PCV and LECs departing the PCV was unchanged at 36 hpf (*Figure 3F,G*). Thus, Yap1 is not required for the induction of lymphatic identity but is needed to establish normal numbers of LECs in sprouts arising from the PCV.

To further investigate the cellular behaviours controlled by Yap1, we performed time-lapse imaging from 32 to 65 hpf. We visualised sprouting LECs using *Tg(lyve1b:dsRed2; fli1a:nlsEGFP)*, which

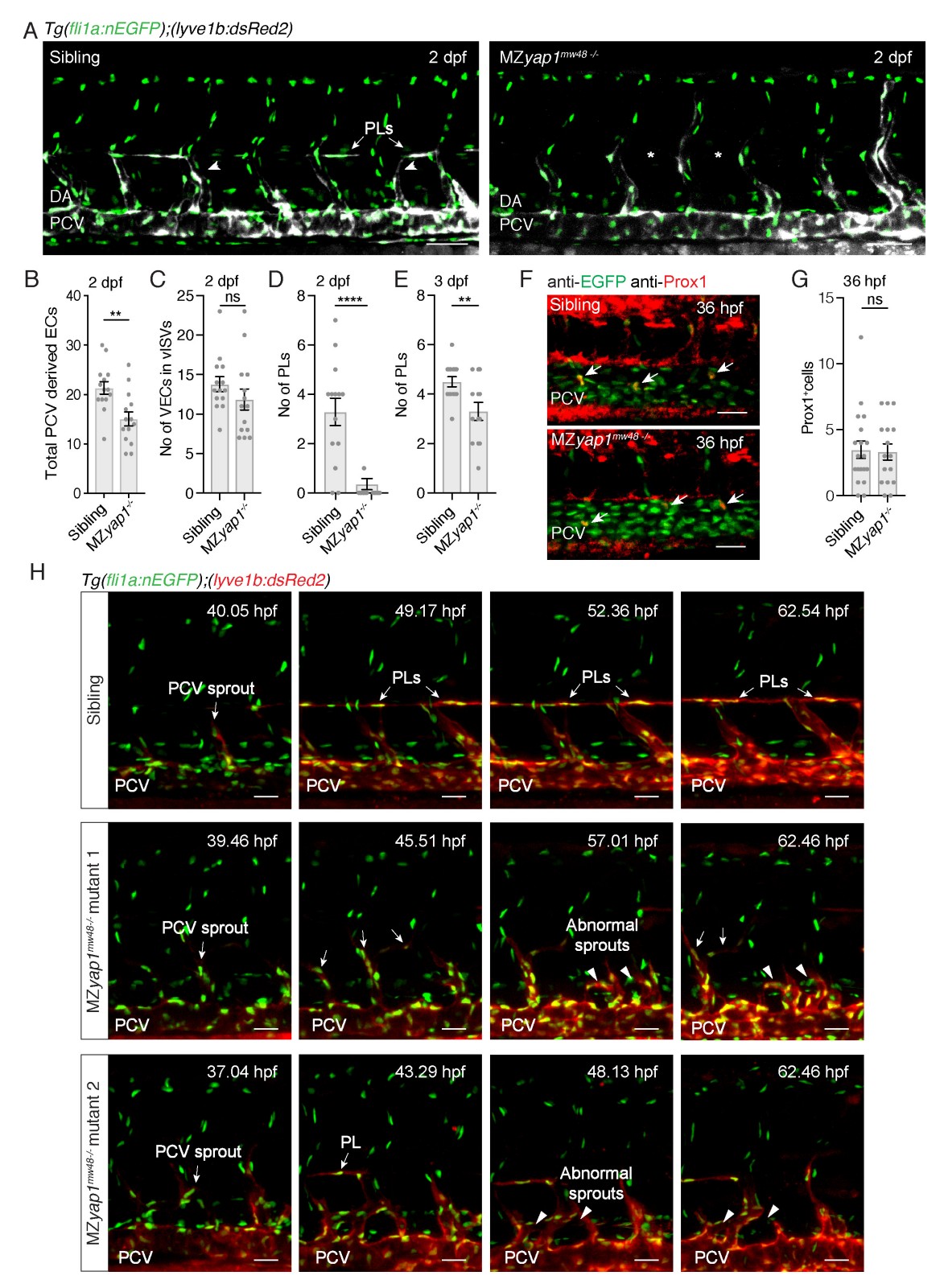

**Figure 3.** MZyap1⁻/⁻ mutants display defects in LEC numbers but not specification. (**A**) Trunk vasculature (EC nuclei in green, veins and lymphatics in white) of sibling and MZyap1^mw48-/- mutant at 2 dpf. Arrowheads indicate posterior cardinal vein (PCV) sprouts. Asterisks mark absent parachordal LECs (PLs). Dorsal aorta (DA). Scale bars: 50 µm. (**B**) Total number of *lyve1*-positive ECs departing the PCV across 6 somites at 2 dpf. (**C**) Number of endothelial cells in venous intersegmental vessels (vISV) across 6 somites at 2 dpf (sibling: 13 ± 0.94, n = 14; MZyap1⁻/⁻: 12 ± 1.33, n = 14; p=0.24 (ns)).
*Figure 3 continued on next page*

*Figure 3 continued*

(**D**) Number of PLs scored across 6 somites at 2 dpf (sibling: 3 ± 0.56, n = 14; MZ*yap1*$^{-/-}$: 0.36 ± 0.23, n = 14; p<0.0001 (\*\*\*\*)). (**E**) Number of PLs scored across 6 somites at 3 dpf (sibling: 4.5 ± 0.20, n = 14; MZ*yap1*$^{-/-}$: 3.31 ± 0.36, n = 13; p=0.0075(\*\*)). (**F**) Immunofluorescence staining for EC nuclei (green) and Prox1 (red) in sibling and MZ*yap1*$^{mw48-/-}$ mutants at 36 hpf. Arrows point to Prox1$^+$ LEC progenitors. Scale bars: 30 μm. (**G**) Quantification of Prox1 +cells in PCV and CV sprouts scored across 6 somites at 36 hpf (sibling: 3.47 ± 0.65, n = 19; MZ*yap1*$^{-/-}$: 3.31 ± 0.61, n = 16; p=0.86 (ns)). (**H**) Maximum projection stills from time-lapse Videos from 32 to 65 hpf. Sibling still images show normal lymphangiogenesis with PCV sprouts, PL formation, sprout detachment and PL proliferation (upper panels). MZ*yap1*$^{mw48-/-}$ mutant one displays abnormal sprouting and looping of PCV sprouts that are retained until the end of the Video (central panels). MZ*yap1*$^{mw48-/-}$ mutant two also exhibits abnormal sprouting and PCV loop formation but also forms PLs (lower panels). Scale bars: 25 μm. Timelapse imaging began at 32 hpf.
DOI: https://doi.org/10.7554/eLife.42881.015

The following source data is available for figure 3:

**Source data 1.** Cell counts for PCV-derived cells in the MZ*yap*$^{-/-}$ mutants.
DOI: https://doi.org/10.7554/eLife.42881.016

at these stages spans the period of first detection of dsRed2 expression (*Figure 3H*, *Videos 4–6*). We observed striking abnormal behaviours of sprouts arising from the PCV in MZ*yap1*$^{-/-}$ mutants, most commonly sprouts appeared thickened, retracted, or turned ventrally and formed abnormal loops (*Video 5*, mutant 1). Often venous sprouts formed abnormal connections with other venous sprouts from adjacent body segments (*Video 6*, mutant 2). These abnormal behaviours and reduced sprouting cell numbers likely contribute to fewer PLs accurately seeding the HM during development.

## Yap1 mediates Vegfc-driven proliferation in venous-derived endothelial cells

YAP is established to control cellular proliferation in diverse contexts, including in ECs (*Panciera et al., 2017*; *Zhao et al., 2011*). To ask whether Yap1 plays a role in EC proliferation downstream of Vegfc in PLs, as well as mediating normal sprouting behaviour, we used a transgenic approach. We overexpressed *vegfc* throughout the trunk under the control of the *prox1a* promoter (using *TgBAC(prox1a: KalTA4-4xUAS-ADV.E1b:TagRFP)*$^{nim5}$, *Tg(10xUAS:vegfc)*$^{uq2bh}$ double transgenic (*vegfc*-OE) embryos), which is active in tissues that include muscle, neurons and vasculature (*Koltowska et al., 2015a*; *van Impel et al., 2014*). We then injected a *yap1* targeting morpholino (MO) that has been previously validated (*Loh et al., 2014*) and phenocopied MZ*yap1*$^{-/-}$ mutants (*Figure 4A,C* and data not shown). *vegfc* overexpression resulted in hyper-proliferation of venous-derived ECs throughout the trunk including prominently in the HM, where excessive numbers of ECs proliferate as previously described (*Figure 4B,C*, *Koltowska et al., 2015a*). The effect of Vegfc stimulation was completely blocked upon *yap1* MO injection but not upon *p53* (control) MO injection (*Figure 4B, C*). Together, these observations are consistent with a role for Yap1 downstream of Vegfc, functioning in Vegfc-induced proliferation.

To further validate this observation made with MO-mediated knockdown, we tested the ability of elevated Vegfc-induced signalling to stimulate PL proliferation in MZ*yap1*$^{-/-}$ mutant embryos. We transplanted cells at blastula stages from *vegfc*-OE transgenic donor embryos into wildtype and MZ*yap1*$^{-/-}$ mutant embryos (*Figure 4D–E*). Embryos that had engrafted *vegfc*-OE cells in muscle adjacent to the HM were then selected for analysis. The number of PLs responding to the local graft were scored

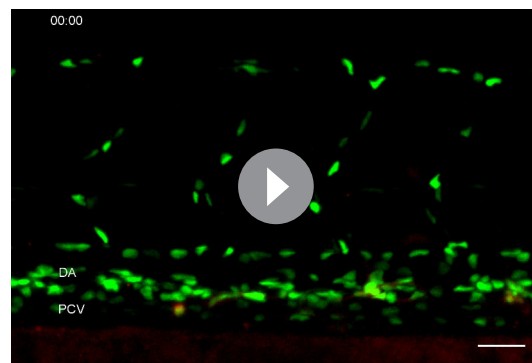

**Video 4.** Time-lapse Video of the trunk vasculature of a sibling from 32 to 65 hpf in the [*Tg(fli1a:nEGFP); (−5.2lyve1b:DsRed2)*]] background showing endothelial nuclei in green and lymphatic and venous endothelium in red. Normal lymphangiogenesis involves sprouting from the posterior cardinal vein (PCV) to form PLs at the horizontal myoseptum. Dorsal aorta (DA). Scale bar: 25 μm.
DOI: https://doi.org/10.7554/eLife.42881.017

within the same body segment in wildtype and MZ*yap1*$^{-/-}$ mutant embryos at 3 dpf. Strikingly, the MZ*yap1*$^{-/-}$ mutant host embryos displayed a vast reduction (although not a complete block) in numbers of proliferative PLs adjacent to HM grafts compared with wildtype siblings (*Figure 4F,G*). These data, coupled with reduced numbers of ECs in sprouts departing the PCV (*Figure 3B*), together suggest that Yap1 plays a role in the proliferation of developing lymphatic progenitors stimulated by Vegfc.

## Vegfc signalling promotes nuclear YAP in LEC progenitors

Nuclear localisation of YAP has been linked to EC proliferation (*Panciera et al., 2017*). We speculated that Vegfc signalling might promote nuclear Yap1 in order to promote LEC proliferation. To test this, we used a transplantation approach and transplanted *vegfc*-overexpressing cells (labelled by TagRFP in host embryos) at blastula stages into embryos carrying the EGFP-YAP reporter (*Figure 5A*). We selected embryos with labelled *vegfc*-OE neurons or muscle cells and found that these grafts were sufficient to induce local proliferation of adjacent venous ECs and LEC progenitors (*Figure 5B,C*). We analysed the localisation of EGFP-YAP in responding PLs local to Vegfc-producing donor cells. To generate an unbiased quantification that allows for variation in transgene intensity between embryos, we scored nuclear EGFP intensity in dorsal aorta cells (DA) and used the average pixel intensity of the DA cells as *vegfc*-OE unresponsive internal controls (see *Koltowska et al., 2015a*). Thus, PL nuclear EGFP intensity is calculated as PL nuclear intensity/average DA nuclear intensity in the same embryo (*Figure 5C',D*). We found that Vegfc responsive PLs displayed an increase in nuclear EGFP-YAP compared with control PLs (*Figure 5C',D*). This increase in nuclear EGFP-YAP in the *vegfc*-OE transplanted embryos was concomitant with a higher number of PLs (*Figure 5G*). Control PLs were scored in embryos transplanted with donor cells expressing the Kalt4 driver but not carrying the UAS:*vegfc* element (*Figure 5B–D,F–G*). Further, we transplanted from EGFP-YAP reporter donor embryos into *vegfc*-OE and control recipient embryos (*Figure 5—figure supplement 1A*). We selected small vascular EC grafts and examined grafted LECs and VECs for EGFP-YAP nuclear concentration. Relative to nuclear concentration in grafted *vegfc*-unresponsive DA cells, EGFP-YAP was increased in nuclei in lymphatic and venous EC grafts in the *vegfc*-OE hosts (*Figure 5—figure supplement 1B–E*).

Finally, as Vegfc/Flt4 signalling has been previously shown to be transduced via Erk in zebrafish trunk vasculature (*Koltowska et al., 2015a*; *Shin et al., 2016*), we investigated the impact of blocking this pathway on EGFP-YAP in PLs. Treatment with SL327, a MEK inhibitor that blocks Erk signalling in zebrafish (*Shin et al., 2016*), decreased the concentration of EGFP-YAP in the nuclei of PLs, concomitant with a reduction in PL number (*Figure 5E,H–J*). This was observed using time-lapse Videos (*Figure 5H*, not shown) as well as acquiring images following a single 12 hr treatment to exclude interference from time-lapse bleaching (*Figure 5I*). Overall, these observations indicate that the Vegfc-Erk axis promotes high nuclear Yap1 and drives normal sprouting and proliferation during zebrafish developmental lymphangiogenesis.

## Discussion

Hippo pathway signalling and YAP/TAZ activity are key regulators of cell proliferation and organ growth and pathway components are frequently dysregulated in cancer (*Harvey et al., 2013*). Recent research has broadened our understanding of this pathway, which can function to integrate stimuli from the cellular environment that ultimately affect cell migration, growth and even cell fate (*Panciera et al., 2017*). In ECs, YAP/TAZ signalling has been relatively understudied, yet has still been reported to have roles in proliferation, survival, migration, adhesion and cellular rearrangements, although the mechanisms that utilise YAP are not completely understood (*Kim et al., 2017*; *Neto et al., 2018*; *Wang et al., 2017*). We show here that loss of Yap1 has a dramatic effect upon lymphatic vascular development, as MZ*yap1*$^{-/-}$ mutants fail to form a lymphatic network in the zebrafish trunk. Surprisingly, these mutants exhibit a largely normal blood vasculature. The severity of the MZ*yap1*$^{-/-}$ mutant phenotype indicates that Yap1 is indispensable for lymphangiogenesis and its paralogue Taz is not able to fully compensate in lymphatic vasculature. While not studied here, it remains possible that Taz may compensate for the absence of Yap1 in some vascular beds (eg. blood vessels) and not others (eg. lymphatics). It is also possible that there are tissue specific effectors of Yap1 in lymphatic endothelial cells by comparison to other vascular lineages. Further work is

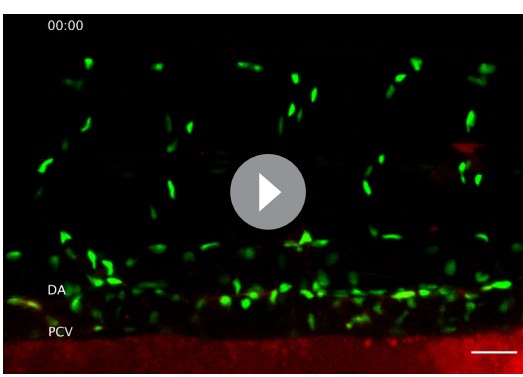

**Video 5.** Time-lapse Video of MZ*yap1*$^{mw48-/-}$ mutant 1 from 32 to 65 hpf with endothelial nuclei in green and lymphatic and venous endothelium in red. Abnormal sprouts emerge from the posterior cardinal vein (PCV). Dorsal aorta (DA). Scale bar: 25 μm.

DOI: https://doi.org/10.7554/eLife.42881.018

clearly needed to determine the mechanisms underlying the restricted lymphatic phenotype that we have observed here.

Lymphatic endothelial progenitor cells are specified in MZ*yap1*$^{-/-}$ mutants but these fail to sprout normally to the horizontal myoseptum. The processes that are necessary for departure of LEC progenitors from the PCV and seeding of the HM include cell sprouting and directional migration, both Vegfc-dependent (*Koltowska et al., 2015a*; *Nicenboim et al., 2015*). Mutants displayed variable and abnormal cell behaviours including turning of sprouts back towards the PCV and fusion of adjacent sprouts. What the precise role of Yap1 is in venous EC sprouting requires further study, yet recent work points to regulation of junctional dynamics and cellular elongation as candidate cellular processes in other vessel contexts (*Neto et al., 2018*).

In addition to sprouting defects, we see a general reduction in the number of ECs that depart the PCV during secondary angiogenesis in the MZ*yap1*$^{-/-}$ mutants. We further show that Vegfc-induced VEC and PL proliferation (in a transgenic overexpression model) requires normal levels of Yap1. In previous studies of blood vascular ECs, the Yap/Taz dependent proliferative response was regulated by Yap/Taz localization in the nucleus, where they regulate transcription to promote cell proliferation (*Neto et al., 2018*; *Sakabe et al., 2017*; *Wang et al., 2017*). Here, we report that EGFP-YAP localises to the nuclei of early LEC progenitors in zebrafish. Moreover, in vivo live imaging showed that dynamic changes occur in EGFP-YAP protein concentration in lymphatic progenitor nuclei. This may indicate active regulation of nuclear-cytoplasmic localisation of Yap1, as has recently been live-imaged in *Drosophila* (*Manning et al., 2018*). Directly imaging nuclear-cytoplasmic shuttling of EGFP-YAP presented challenges, chiefly the highly variable morphology of PL cytoplasm, which limited us to calculations and image analysis in 2D and may have introduced technical variation. Nevertheless, nuclear/cytoplasmic EGFP-YAP measurements correlated with changes in nuclear concentration of EGFP-YAP between

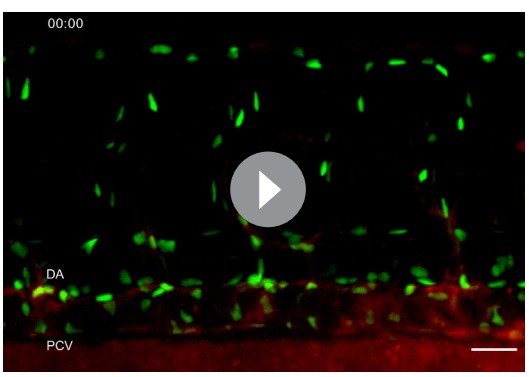

**Video 6.** Time-lapse Video of the trunk vasculature in MZ*yap1*$^{mw48-/-}$ mutant 2 from 32 to 65 hpf, showing endothelial nuclei in green and lymphatic and venous endothelium in red. Abnormal sprouts emerge from the posterior cardinal vein (PCV) anddisplaying loops as well as fusions with adjacent sprouts. Dorsal aorta (DA). Scale bar: 25 μm.

DOI: https://doi.org/10.7554/eLife.42881.019

PLs (*Figure 1* and *Figure 1—figure supplement 1*) and confirmed that PLs display highly variable concentrations of nuclear EGFP-YAP that likely reflect differences in Yap1 activation and upstream signalling. We also note that we did not observe enrichment of EGFP-YAP fusion protein at cell-cell junctions and we note a caveat that YAP may also be regulated at the level of protein turnover. Importantly, a high concentration of nuclear EGFP-YAP was promoted by a local source of Vegfc, concomitant with increased PL proliferation (*Figure 5* and *Figure 5—figure supplement 1*). Inhibition of Flt4-dependent Erk signalling also reduced the nuclear concentration of EGFP-YAP (*Figure 5E–J*). This suggests overall that paracrine Vegfc signalling activates LEC progenitor Flt4/Erk-signalling and that Yap1 is an essential effector of this pathway.

The development of the lymphatic vasculature is a remarkable process whereby a limited pool of progenitor cells in the wall of functional

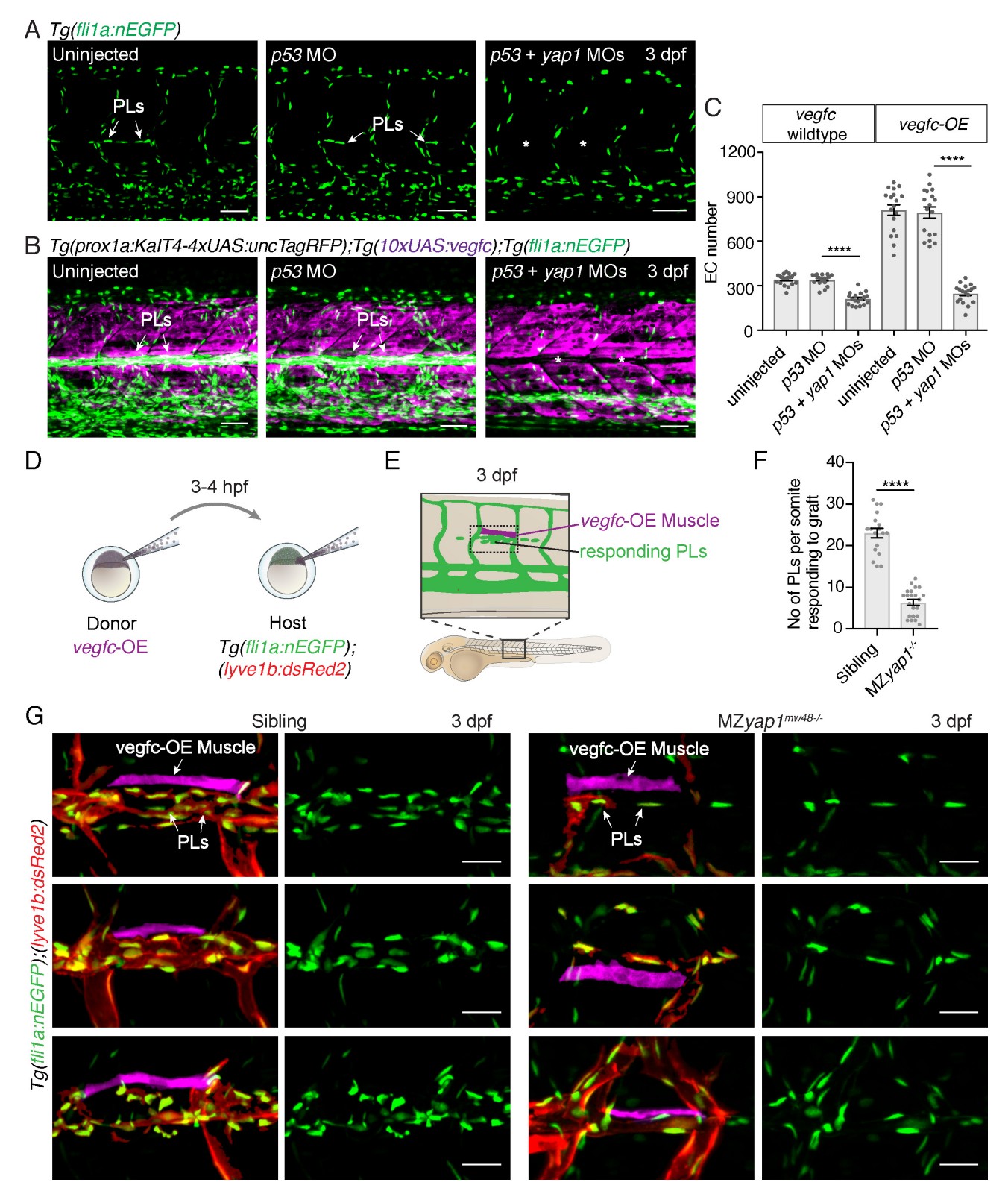

**Figure 4.** Yap1 mediates endothelial cell proliferation downstream of Vegfc. (**A**) Trunk vasculature at 3 dpf displaying endothelial cells (ECs) in green from uninjected, *p53* morpholino (MO) injected and *p53 +yap1* MO injected embryos. Asterisks highlight missing parachordal LECs (PLs). Scale bars: 50 µm. (**B**) Trunk vasculature at 3 dpf from uninjected, *p53* MO injected and *p53 +yap1* MOs injected embryos of the *Tg(prox1a:KalT4xUAS:uncTagRFP); Tg(10xUAS:vegfc)* strain. *yap1* MO injection rescues the EC proliferation phenotype. ECs in green. Asterisks mark missing PLs. Scale bars: 50 µm. (**C**)
*Figure 4 continued on next page*

*Figure 4 continued*

Quantification of total EC number across four somites in *vegfc*-unstimulated (uninjected: 343 ± 9, n = 18; *p53* MO: 341 ± 9, n = 18; *p53* +*yap1* MOs: 248 ± 15, n = 18; p<0.0001(****)) and *vegfc*-stimulated embryos (uninjected: 811 ± 35, n = 18; *p53* MO: 794 ± 38, n = 18; *p53* +*yap1* MOs: 212 ± 10, n = 18; p<0.0001(****)). (D) Schematic representation of *vegfc*-overexpressing cell transplantations for single muscle grafts in [*Tg(fli1a:nEGFP); (−5.2lyve1b:DsRed2)*] hosts of siblings and MZ*yap1*$^{mw48-/-}$ mutants. (E) Schematic of transplanted host embryo at 3 dpf. Muscle grafts produce excessive Vegfc causing a hyperproliferation response in adjacent PLs. (F) Quantification of PL number within one somite responding to the *vegfc*-OE single muscle graft at 3 dpf (sibling: 23.00 ± 1.19, n = 18; MZ*yap1*$^{mw48-/-}$: 6.36 ± 0.71, n = 22; p<0.0001 (****)). (G) Examples of 3 different *vegfc*-OE muscle grafts (false coloured) in siblings (left panels) and MZ*yap1*$^{mw48-/-}$ mutants (right panels). Merge images show ECs in green, lymphatic and venous ECs are red (yellow) [*Tg(fli1a:nEGFP);(−5.2lyve1b:DsRed2)*]. Scale bars: 25 μm.

DOI: https://doi.org/10.7554/eLife.42881.020

The following source data is available for figure 4:

**Source data 1.** Measurements of ECs in Vegfc-overexpression embryos.

DOI: https://doi.org/10.7554/eLife.42881.021

embryonic blood vessels are utilised as progenitors to generate a complex second vasculature. This inherently presents a tissue growth event that requires extensive, coordinated cellular proliferation coincident with network morphogenesis. That the HIPPO pathway and YAP would play a role follows a clear biological rationale. The process of lymphangiogenesis is important in a swathe of disease settings that include in common cardiovascular diseases and in the metastatic spread of cancer (*Petrova and Koh, 2018*). In the later setting, excessive lymphangiogenesis underpins metastatic spread and is a target of interest for inhibition (*Dieterich and Detmar, 2016*). Excessive lymphangiogenesis is also a target of interest in rare diseases such as lymphangioma or lymphatic malformation. While much emphasis in the development of anti-lymphangiogenic agents has focused on targeting VEGFC/VEGFR3 signalling, pathways involved in tissue growth and cellular proliferation such as the Hippo pathway, YAP and TAZ may well represent alternative targets worthy of further investigation.

# Materials and methods

## Key resources table

| Reagent type (species) or resource | Designation | Source or reference | Identifiers | Additional information/ reagent source |
|---|---|---|---|---|
| Genetic reagent (*D.rerio*) | *yap1*$^{mw48-/-}$ | *Miesfeld et al., 2015* | RRID:ZFIN_ ZDB-ALT-160122-5 | Brian Link (Medical College of Wisconsin, Milwaukee, USA) |
| Genetic reagent (*D.rerio*) | *yap1*$^{ncv101-/-}$ | *Nakajima et al., 2017* | RRID:ZFIN_ ZDB-ALT-170522-16 | Naoki Mochizuki (National Cerebral and Cardiovascular Centre Research Institute, Suita, Osaka) |
| Genetic reagent (*D.rerio*) | *Tg(fli1a:nEGFP)*$^{y7}$ | *Lawson et al., 2002* | RRID:ZFIN_ ZDB-ALT-060821-4 | Brant M Weinstein (National Institute of Child Health and Human Development, Bethesda, USA) |
| Genetic reagent (*D.rerio*) | *Tg(- 5.2lyve1b:DsRed)*$^{nz101}$ | *Okuda et al., 2012* | RRID:ZFIN_ ZDB-ALT-120723-3 | Phil and Kathy Crosier (Department of Molecular Medicine University of Auckland School of Medicine) |
| Genetic reagent (*D.rerio*) | *Tg(fli1:EGFP-YAP)*$^{ncv35}$ | *Nakajima et al., 2017* | RRID:ZFIN_ ZDB-ALT-170522-18 | Naoki Mochizuki (National Cerebral and Cardiovascular Centre Research Institute, Suita, Osaka) |

*Continued on next page*

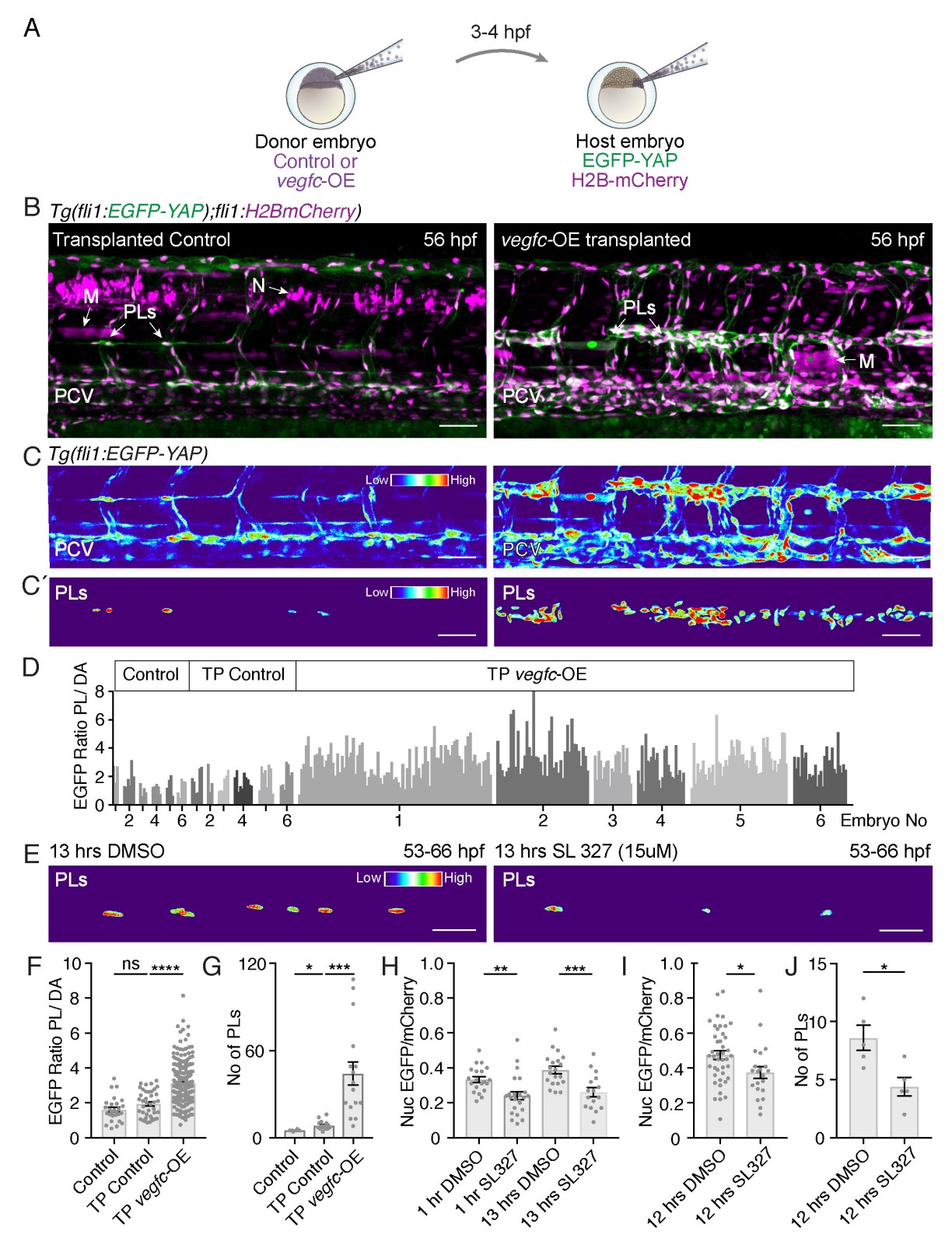

**Figure 5.** Vegfc promotes nuclear Yap1 in developing lymphatic progenitors. (**A**) Schematic showing the transplantation of *vegfc*-OE cells into EGFP-YAP reporter hosts. (**B**) Trunk vasculature of YAP reporter host embryos with neuron and muscle grafts expressing RFP (magenta) at 56 hpf. Control grafts (left) and *vegfc-OE* grafts (right). Posterior cardinal vein (PCV); Muscle (**M**); parachordal LECs (PLs); Neuron (**N**). Scale bars: 50 µm in B and C. (**C**) Heatmaps of maximum projections of EGFP-YAP from B) showing PLs and PCV. Red corresponds to high EGFP-YAP fluorescence. (**C'**) Lower panels

*Figure 5 continued on next page*

*Figure 5 continued*

show heatmaps of EGFP-YAP in PL nuclei. (D) EGFP intensity for each PL expressed as a ratio to the average of 6 dorsal aorta (DA) cells (unresponsive to *vegfc*). six embryos per group: untransplanted host (control) (PL n = 27), transplanted control without the *UAS:vegfc* construct (TP Control) (PL n = 40) and *vegfc*-OE transplanted embryos (TP *vegfc*-OE) (PL n = 251). Individual embryos indicated by an individual grey shade. (E) Heatmaps of EGFP-YAP depicting PL nuclei of embryos at 66 hpf that were treated with DMSO (left) and SL327 for 13 hr (hrs) (right). Max projections are from time-lapse Videos. Scale bars: 30 μM. (F) EGFP mean intensity for each PL/DA in panel D (Control: 1.61 ± 0.13, n = 27; TP Control: 1.93 ± 0.12, n = 40; p=0.08 (ns); TP *vegfc*-OE: 3.14 ± 0.07, n = 251; p<0.0001(****)). (G) Quantification of PL number across six somites for each group (Control: 5 ± 0.26, n = 6; TP Control: 8 ± 0.78, n = 16; p=0.0157 (*); TP *vegfc*-OE: 44 ± 8.04, n = 16; p=0.0001(***)). (H) Quantifications of the mean EGFP/mCherry fluorescent intensity ratio in PL nuclei from time-lapse Video stills. Embryos were mounted in DMSO as control (n = 5) or SL327 (15 μM) (n = 5) and continuously imaged for 12 hr starting at 54 hpf (1 hr drug treatment), finishing at 66 hpf (13 hr) (1 hr DMSO: 0.33 ± 0.02, n = 20; 1 hr SL327: 0.24 ± 0.02, n = 23; p=0.0036 (**)) (13 hr DMSO: 0.39 ± 0.02, n = 20; 13 hr SL327: 0.26 ± 0.03, n = 16; p=0.0006(***)). (I) Quantifications of mean EGFP/mCherry Ratio in PLs of embryos at 66 hpf treated with DMSO/SL327 for 12 hr from 54 hpf to 66 hpf without time-lapse imaging (12 hr DMSO: 0.47 ± 0.02, PLs n = 43; 13 hr SL327: 0.37 ± 0.03, PLs n = 22; p=0.02 (*)). (J) Quantification of PL number in embryos treated with DMSO/SL327 for 12 hr at 66 hpf without time-lapse imaging (12 hr DMSO: 8.6 ± 1.08, embryo n = 5; 13 hr SL327: 4.4 ± 0.81, embryo n = 5; p=0.014 (*)).

DOI: https://doi.org/10.7554/eLife.42881.022

The following source data and figure supplements are available for figure 5:

**Source data 1.** Measurements of LEC EGFP-YAP in response to Vegfc.

DOI: https://doi.org/10.7554/eLife.42881.025

**Figure supplement 1.** The autonomous EGFP-YAP response to Vegfc in transplanted ECs.

DOI: https://doi.org/10.7554/eLife.42881.023

**Figure supplement 1—source data 1.** Measurements of EGFP-YAP PLs in Vegfc-overexpression embryos.

DOI: https://doi.org/10.7554/eLife.42881.024

*Continued*

| Reagent type (species) or resource | Designation | Source or reference | Identifiers | Additional information/ reagent source |
|---|---|---|---|---|
| Genetic reagent (*D.rerio*) | Tg(fli1:Gal4db-TEAD2ΔN-2A-mC)^ncv36 | *Nakajima et al., 2017* | RRID:ZFIN_ZDB-ALT-170522-19 | Naoki Mochizuki (National Cerebral and Cardiovascular Centre Research Institute, Suita, Osaka) |
| Genetic reagent (*D.rerio*) | Tg(UAS:GFP)^y1 | *Asakawa et al., 2008* | RRID:ZFIN_ZDB-ALT-011017-8 | Naoki Mochizuki (National Cerebral and Cardiovascular Centre Research Institute, Suita, Osaka). |
| Genetic reagent (*D.rerio*) | Tg(fli1:Myr-mC)^ncv1 | *Kwon et al., 2013* | ZFIN ID: ZDB-FIG-150115–34 | Naoki Mochizuki (National Cerebral and Cardiovascular Centre Research Institute, Suita, Osaka) |
| Genetic reagent (*D.rerio*) | Tg(fli1:H2B-mC)^ncv31 | *Yokota et al., 2015* | RRID:ZFIN_ZDB-ALT-160323–6 | Naoki Mochizuki (National Cerebral and Cardiovascular Centre Research Institute, Suita, Osaka) |
| Genetic reagent (*D.rerio*) | Tg(gata1:DsRed)^sd2 | *Traver et al., 2003* | RRID:ZFIN_ZDB-ALT-051223-6 | Leonard I Zon (Howard Hughes Medical Institute, Boston, MA), European Zebrafish Resource Center (EZRC), Zebrafish International Resource Center (ZIRC). |
| Genetic reagent (*D.rerio*) | TgBAC(prox1a:KalTA4-4xUAS-ADV.E1b:TagRFP)^nim5 | *Dunworth et al., 2014; van Impel et al., 2014* | RRID:ZFIN_ZDB-ALT-160323–6 | Elke Ober (The Danish Stem Cell Centre (DanStem) University of Copenhagen) |

*Continued on next page*

*Continued*

| Reagent type (species) or resource | Designation | Source or reference | Identifiers | Additional information/ reagent source |
|---|---|---|---|---|
| Genetic reagent (D.rerio) | *Tg(10xUAS: vegfc)$^{uq2bh}$* | *Koltowska et al., 2015a* | RRID:ZFIN_ZDB-ALT-151208-1 | Ben M Hogan (Institute for Molecular Bioscience, The University of Queensland) |
| Genetic reagent (D.rerio) | *Tg(fli1a:H2B-mCherry)$^{uq37bh}$* | this paper | | Ben M Hogan (Institute for Molecular Bioscience, The University of Queensland) |
| Genetic reagent (D.rerio) | *TgBAC(dab2b: EGFP)$^{ncv67}$* | other | | Shigetomo Fukuhara and Shinya Yuge (Nippon Med. School) |
| Genetic reagent (D.rerio) | *Tg(kdrl: mCherry)$^{ncv502}$* | other | | Naoki Mochizuki (National Cerebral and Cardiovascular Centre Research Institute, Suita, Osaka). |
| Antibody | chicken anti-GFP | Abcam | Cat#ab13970 | Primary AB, Alexa Fluor-488 conjugated, 1:250 |
| Antibody | C1 anti-GFP | Invitrogen | Cat#A11039 | Secondary AB, 1:400 |
| Antibody | rabbit anti-Prox1 | AngioBio | Cat#11–002 | Primary AB, 1:500 |
| Antibody | anti-rabbit IgG-HRP | Cell Signaling | Cat#7074S | Secondary AB, 1:1000 |
| Sequence-based reagent | *MO4-yap1* | *Loh et al., 2014* | ZFIN ID: ZDB-MRPHLNO-140915–5 | Genetools, LLC |
| Sequence-based reagent | *p53 MO* | *Robu et al., 2007* | SKU: PCO-ZebrafishP53-100 | Genetools, LLC |
| Commercial assay or kit | TSA Plus Cyanine 3 System | Perkin Elmer | #NEL744001KT | Amplification of signal detection of the Prox1-AB staining |
| Chemical compound, drug | SL327 (MEK inhibitor) | Sigma-Aldrich | S4069; CAS:305350-87-2 | 15 mM |
| Software, algorithm | ImageJ | ImageJ (http://imagej.nih.gov/ij/) | SCR:002285 | Image processing and analysis, Version 2.0.0-rc-49/1.51d |
| Software, algorithm | Imaris x64 | Bitplane | SCR:007370 | Image processing and analysis, Version 9.0.2 |
| Software, algorithm | GraphPad Prism | GraphPad Prism (https://graphpad.com) | SCR:015807 | Statistics, Prism7: Version 7.0 c and Prism8: Version 8.0.1 |

## Zebrafish

All zebrafish work was conducted in accordance with the guidelines of the animal ethic committee guidelines at the University of Queensland and of the National Cerebral and Cardiovascular Center (No.14005 and No.15010). The two different mutant strains used were the *yap1$^{ncv101-/-}$* mutant, which has a 25 base pair (bp) deletion in exon1 leading to a frame shift and stop codon after 71 bp, prior to the TEAD binding domain (*Nakajima et al., 2017*), and the *yap1$^{mw48-/-}$* mutant, which displays a 4 bp deletion after 158 bp, truncating the TEAD binding domain and leading to a premature stop codon (*Miesfeld et al., 2015*). The transgenic zebrafish lines used were published previously as following: *Tg(fli1a:nEGFP)$^{y7}$* (*Lawson et al., 2002*), *Tg(- 5.2lyve1b:DsRed)$^{nz101}$* (*Okuda et al., 2012*), *Tg(fli1:EGFP-YAP)$^{ncv35}$* and *Tg(fli1:Gal4db-TEAD2ΔN-2A-mC)$^{ncv36}$* (*Nakajima et al., 2017*), *Tg(UAS:*

GFP)$^{y1}$ (**Asakawa et al., 2008**), Tg(fli1:Myr-mC)$^{ncv1}$ (**Kwon et al., 2013**) and Tg(fli1:H2B-mC)$^{ncv31}$ (**Yokota et al., 2015**), Tg(gata1:DsRed)$^{sd2}$ (**Traver et al., 2003**), TgBAC(prox1a: KalTA4-4xUAS-ADV.E1b:TagRFP)$^{nim5}$(**Dunworth et al., 2014**; **van Impel et al., 2014**); Tg(10xUAS:vegfc)$^{uq2bh}$ (**Koltowska et al., 2015b**). The Tg(fli1a:H2B-mCherry)$^{uq37bh}$ strain was generated for this study using Gateway cloning and transgenesis. The transgenic line TgBAC(dab2b:EGFP)$^{ncv67}$ was provided by Shigetomo Fukuhara and Shinya Yuge (Nippon Med. School). Tg(kdrl:mCherry)$^{ncv502}$ were generated using Tol2 mediated transgenesis.

## Morpholino injections

The yap1 morpholino (MO) used has been validated and described previously as MO4-yap1 (**Loh et al., 2014**). To control for yap1 morpholino non-specific effects, we co-injected p53 MO as previously described (**Robu et al., 2007**) and both uninjected and p53 MO injected embryos were used as controls. All MOs were purchased from Genetools, LLC and injected at 5 ng/embryo.

### Genotyping of yap1 mutants

Yap1 mutants of the yap1$^{ncv101}$ allele were genotyped by PCR using a Microchip Electrophoresis System for DNA/RNA Analysis (MCE-202 MultiNA). yap1$^{ncv101}$ forward primer: TCCTTCGCAAGGC TTGGATAATTG yap1$^{ncv101}$ reverse primer: TTGTCTGGAGTGGGACTTTGGCTC yap1 mutants carrying the yap1$^{mw48}$ allele were genotypes using a KASP genotyping assay following the manufacturer's instructions (LGC Genomics).

## Drug treatments and immunohistochemistry

Embryos were treated with 15 µM of the chemical inhibitor SL327 (Merk, NJ, USA) diluted in E3 medium with 0.003% 1-phenyl-2-thiourea (PTU) and 1% DMSO (Sigma) and immobilised with Tricaine (0.08 mg/ml) and 1% low melting agarose for imaging. Control embryos were kept in E3-PTU water with 1% DMSO.

Immunofluorescent staining was performed as described in **Okuda et al. (2018)** with the following minor changes to the protocol: 30 min of ProtK treatment (20 ng/ml) was used for 36 hpf old embryos. For EGFP, the primary antibody used was chicken a-GFP (1:400, Abcam, #ab13970) and secondary C1 anti-GFP (Invitrogen, #A11039). For Prox1, rabbit a-Prox1 (1:500, AngioBio, #11–002) was used as primary and a-rabbit IgG-HRP (1:1,000, Cell Signaling, #7074S) as secondary and amplified the Prox1 signal with TSA Plus Cyanine 3 System (Perkin Elmer, #NEL744001KT).

## Transplantations

EC transplantations to test cell autonomy were performed between pre-dome stage donor and host embryos essentially as previously published (**Hogan et al., 2009a**). Briefly, only successfully transplanted embryos with a normal morphology were selected to be imaged. Despite each transplanted graft being unique, comparable EC grafts were selected in terms of location and size for each experiment. In **Figure 2G–I**, the EC graft size was categorized as small, medium or large to keep grafts comparable between wildtype and mutant embryos studied. 'Large' grafts spanned vasculature over 4–5 somites, 'medium' sized grafts spanned vasculature over 2–3 somites and 'small' grafts over 1–2 somites. Single cell grafts were not included in the analysis. To analyse the response of PLs to locally transplanted vegfc-overexpressing muscle cell grafts in **Figure 4**, only single cell muscle grafts adjacent to the horizontal myoseptum were considered which spanned one somite. The PL number was quantified within the same somite. In **Figure 5—figure supplement 1**, selected host embryos showed lymphatic or venous ECs as well as transplanted ECs in the dorsal aorta.

All EC graft donor embryos were in-crosses of either wildtype control or homozygous Zyap1$^{ncv101}$ mutants. For the transplantation of muscle and neurons in **Figure 5**, donor embryos for the chosen transgenes [TgBAC(prox1a: KalTA4-4xUAS-ADV.E1b:TagRFP)] as TP control and [TgBAC(prox1a: KalTA4-4xUAS-ADV.E1b:TagRFP);Tg(10xUAS:vegfc)]as vegfc-OE were all heterozygous. Crosses between Tg(fli1:EGFP-YAP)$^{ncv35}$ and Tg(fli1a:H2B-mCherry)$^{uq37bh}$ were used to generate heterozygous host embryos.

## Imaging

All imaged embryos were immobilised with tricaine (0.08 mg/ml) and mounted laterally in 1% low-melting agarose (Sigma-Aldrich, A9414-100G) for still images and 0.7% agarose for time-lapse Videos. Zebrafish embryos were imaged at the Australian Cancer Research Foundation's Cancer Ultrastructure and Function Facility at the Institute for Molecular Bioscience in Brisbane, Australia using a Zeiss LSM 710 FCS confocal microscope and an Andor Dragonfly Spinning Disc Confocal microscope with the Zyla 4.2 sCMOS camera (exclusively for *Video 2* and *Figure 1—figure supplement 1F*), or at the National Cerebral and Cardiovascular Center Research Institute in Osaka, Japan on an OLYMPUS confocal microscope (FluoView FV1000 and FV1200). All embryos analysed for quantification of signal intensity were heterozygous carriers for each transgene and imaged with the same imaging settings for each experiment with neuronal and muscle transplantations being the exception.

## Image processing and fluorescent intensity analysis

Images were processed with image processing software ImageJ Version 2.0.0-rc-49/1.51d (National Institute of Health) and Imaris x64 (Version 9.0.2). For *Figure 2B*, the dorsal longitudinal lymphatic vessel (DLLV) could not be scored due to interference with skin fluorescence in the *TgBAC(dab2b: EGFP)* strain.

To investigate EGFP-YAP activity within ECs, we used three different approaches. The first method, which we chose to use throughout the study examined nuclear EGFP/mCherry intensity ratios within ECs. This analysis was performed using Imaris software to extract the average pixel intensity spanning the entire nucleus in 3D. The nucleus was masked using the red channel (mCherry). To represent both fluorophores in one graph for each cell track shown in *Figure 1I* and *Figure 1—figure supplement 1G*, the mean mCherry intensity was used to normalize the graphs. In the second method, nuclear EGFP/mCherry intensity was calculated manually in 2D on a single z-plane using ImageJ. The chosen z-plane was determined by the centre of each PL nucleus. Intensities were extracted by thresholding the red channel. The final method calculated the nuclear to cytoplasm ratio of EGFP-YAP using the same manual approach with ImageJ with the addition that the cell outline is manually drawn around each cell and the all values within the thresholded nucleus are excluded.

For *Figure 5*, measurements of relative EGFP-YAP intensity, to control for the variation in transgene signal between embryos, the average EGFP fluorescent intensity per pixel for individual PLs was measured and divided by the mean of 6 DA cells measured within the same embryo. A similar approach was used for *Figure 5—figure supplement 1* with the modification that at least 3 DA cells were averaged due to graft size limitations.

*Figure 5C' and E*, *Videos 2–3* and *Figure 5—figure supplement 1C"-D"* display the EGFP fluorescence intensity as a heatmap within the PL nuclei which was generated using the Imaris surface function for the red channel to select only the nuclei.

## Quantification and statistical analysis

Graphic representation of data and statistical analysis was performed using Prism7 and Prism8 (GraphPad, version 7.0 c for Prism7 and version 8.0.1 for Prism8). All experiments were statistically evaluated using the unpaired two-sided t-test as comparison of mean unless stated otherwise. Stars indicate p-values as level of significance in GP style with $p \leq 0.0001$ (****), $p \leq 0.0002$ (***), $p \leq 0.0021$ (**), $p \leq 0.332$ (*), and $p \geq 0.05$ (not significant (ns)). All graphs with single dot scatter plots with error bars showing mean and standard error of the mean (SEM). A pearson correlation analysis and linear regression has been performed in *Figure 1H* and *Figure 1—figure supplement 1C* to compare the similarity of the different methods to measure the average EGFP intensity.

The EC numbers in *Figure 4C* and *Figure 5G and J* were counted using the Spot detection algorithm from Imaris. All other EC numbers were manually counted with ImageJ. The fragments of thoracic duct in *Figure 2D* and the PL abundance in *Figure 3D* were scored as present (1) or absent (0) for each somite per embryo and displayed as percentage.

## Acknowledgements

This work was supported in part by a National Health and Medical Research Council (NHMRC) Project Grant (1079670) and the University of Queensland's Centre for Cardiac and Vascular Biology. BMH was supported by a National Heart Foundation/NHMRC Career Development Fellowship (1083811) and NHMRC Senior Research Fellowship (1155221). We thank Tevin Chau and Nick Condon for technical assistance and useful suggestions. The TgBAC(*dab2:EGFP*) line used in this manuscript was provided by Shigetomo Fukuhara and Shinya Yuge (Nippon Med. School). Imaging was performed in the Australian Cancer Research Foundation's Cancer Ultrastructure and Function Facility at IMB and the imaging centre at the National Cerebral and Cardiovascular Centre, Osaka.

## Additional information

### Funding

| Funder | Grant reference number | Author |
|---|---|---|
| National Heart Foundation of Australia | 1083811 | Benjamin M Hogan |
| National Health and Medical Research Council | 1155221 | Benjamin M Hogan |

The funders had no role in study design, data collection and interpretation, or the decision to submit the work for publication.

### Author contributions

Lin Grimm, Conceptualization, Formal analysis, Investigation, Methodology, Writing—original draft; Hiroyuki Nakajima, Smrita Chaudhury, Formal analysis, Methodology; Neil I Bower, Conceptualization, Resources, Supervision; Kazuhide S Okuda, Investigation, Methodology; Andrew G Cox, Resources, Methodology; Natasha L Harvey, Conceptualization, Resources, Funding acquisition; Katarzyna Koltowska, Conceptualization, Supervision, Methodology; Naoki Mochizuki, Conceptualization, Resources, Supervision, Funding acquisition; Benjamin M Hogan, Conceptualization, Resources, Supervision, Funding acquisition, Writing—review and editing

### Author ORCIDs

Lin Grimm http://orcid.org/0000-0001-7807-2096
Andrew G Cox http://orcid.org/0000-0003-4189-9422
Naoki Mochizuki http://orcid.org/0000-0002-3938-9602
Benjamin M Hogan http://orcid.org/0000-0002-0651-7065

### Ethics

Animal experimentation: All zebrafish work was conducted in accordance with the guidelines of the animal ethic committee guidelines at the University of Queensland and of the National Cerebral and Cardiovascular Center (No.14005 and No.15010).

### Decision letter and Author response

Decision letter https://doi.org/10.7554/eLife.42881.028
Author response https://doi.org/10.7554/eLife.42881.029

## Additional files

### Supplementary files

• Transparent reporting form
DOI: https://doi.org/10.7554/eLife.42881.026

### Data availability

All data generated or analysed during this study are included in the manuscript and supporting files.

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
