## [Decision Letter]

Thank you for submitting your article "Yap promotes sprouting and proliferation of lymphatic progenitors downstream of Vegfc in the zebrafish trunk" for consideration by *eLife*. Your article has been reviewed by three peer reviewers, including Holger Gerhardt as the Reviewing Editor and Reviewer #1, and the evaluation has been overseen by Didier Stainier as the Senior Editor.

The reviewers have discussed the reviews with one another and the Reviewing Editor has drafted this decision to help you prepare a revised submission.

Summary:

This work defines a key role for Yap1 in driving the proliferation of lymphatic endothelial cells and expansion of the lymphatic vascular network downstream of the Vegfc-Vegfr3 pathway. In particular, it demonstrates that Vegfc drives nuclear localization of YAP and that maternal and zygotic mutation of *yap1* severely perturbs lymphatic network formation in zebrafish embryos. Although a role for Yap in driving vascular endothelial cell proliferation and migration is well established, this work is the first to implicate Yap in the control lymphangiogenesis and reveals a critical dependence on Yap for transduction of pro-proliferative Vegfc signaling. Careful analysis including transplantation and overexpression of Vegfc demonstrate that YAP1 is dispensable for early lymphatic specification, but required for Vegfc induced proliferation. Surprisingly, venous sprouting and proliferation appeared largely unaffected, despite the fact that these sprouts also respond primarily to Vegfc. Finally, this work demonstrates that Erk signalling is required for YAP activation suggesting that Vegfc signals through its canonical Erk pathway to drive YAP1 dependent proliferation of lymphatic endothelial cells.

Essential revisions:

The reviewers all find the work very interesting and novel, but concur on three key points that need further clarification before publication. However the reviewers believe this should not need to involve extensive new experiments:

1) Selectivity of the findings for lymphatics: Please clarify how YAP selectively drives VEGF-C dependent proliferation in LECs but not BECs. This should partly be possible by reevaluation of existing videos, including assessment of LEC migration. Specifically, please comment on: is Vegfc-dependent Yap function specific to the lymphatic vasculature in zebrafish and entirely a consequence of reduced proliferation of Prox1 +ve progenitors? This needs further investigation of (i) any earlier defects in blood vascular development, (ii) origins of the observed lymphatic defects in mutant/transplantation experiments (i.e. via reduced specification, migration or proliferation of Prox1 +ve progenitors) and (iii) the cause of later recovery of the lymphatic phenotype (is this a proliferation delay rather than a block)

2) Further evidence, morpholino versus genetic mutant: Evidence that Vegfc promotes proliferation via Yap1 relies on knockdown of *yap1* using a morpholino – and genetic evidence for this important interaction is missing. If mutant lines are not available in the prox1a:KalT4-UAS/UAS:vegfc background, this could be provided using the same transplant approach as in Figure 4 – i.e., upon transplantation of Vegfc-overexpressing cells into control and *yap1* mutant embryos and quantification of a resulting increase (or not) of lymphatic cell numbers, as in Figure 4G.

3) Method and implications of nuclear YAP-EGFP quantifications: Live imaging analyses of EGFP-YAP nuclear accumulation aim to indicate dynamic activation of YAP in lymphatic endothelial cells, but cannot rule out that underlying fluctuations in EGFP-YAP expression may account for observed changes. Transplantation experiments to generate mosaic EGFP-YAP-expressing embryos or mosaic transient expression of EGFP-YAP using Tol2 or likewise would enable dynamic quantification of cytoplasmic to nuclear ratios of single cells and confirm YAP activation dynamics. If these ratios match the nuclear EGFP-YAP to H2B-mCherry ratios for individual cells, then this would provide convincing evidence that further quantification used for Vegfc-stimulation experiments in Figure 4 are appropriate.

Along the same inquiry line, the reviewers point out: In Figure 1E' and Figure 4C' and E EGFP-Yap intensity appears to be inhomogeniously distributed over the nucleus delineated by H2B-mCherry. How does that change in z and wouldn't that observation impact on the quantitative evaluation based on a single confocal plane as described for Figure 1D-E.

In Figure 4H and I, identical treatment conditions have been evaluated after continuous imaging (H) or by acquisition of a single stack, why? If phototoxicity is addressed, please comment in the text.

---

## [Author Response]

Essential revisions:The reviewers all find the work very interesting and novel, but concur on three key points that need further clarification before publication. However the reviewers believe this should not need to involve extensive new experiments:1) Selectivity of the findings for lymphatics: Please clarify how YAP selectively drives VEGF-C dependent proliferation in LECs but not BECs. This should partly be possible by reevaluation of existing videos, including assessment of LEC migration. Specifically, please comment on: is Vegfc-dependent Yap function specific to the lymphatic vasculature in zebrafish and entirely a consequence of reduced proliferation of Prox1 +ve progenitors? This needs further investigation of (i) any earlier defects in blood vascular development, (ii) origins of the observed lymphatic defects in mutant/transplantation experiments (i.e. via reduced specification, migration or proliferation of Prox1 +ve progenitors) and (iii) the cause of later recovery of the lymphatic phenotype (is this a proliferation delay rather than a block)

We agree with the reviewers that further understanding of the earliest phenotypes are needed.

In response to the above points, we have now included additional data in the revised manuscript. While we find that specification is normal in MZ*yap1* mutants at 36 hpf (Figure 3F-G). The number of PLs that make it to the horizontal myoseptum is reduced. We now provide 3 new videos which are representative of a larger set of time-lapse videos and shown as stills in revised Figure 3.

This analysis shows that the reduction in PLs reaching the horizontal myoseptum (HM) occurs concomitantly with abnormal sprouting behaviours during secondary angiogenesis. We consistently observe that venous sprouts display abnormal morphological features, loss of directionality, often fuse with sprouts from adjacent segments and that fewer PLs are able to reach the HM during the early stages of secondary sprouting. prox1a:KalT4-UAS/UAS:vegfc While it is difficult to quantify these diverse aberrant cellular behaviours, we believe that the videos and still images in Figure 3 address the reviewer’s major questions above.

We also provide careful quantification of EC numbers for: all Lyve1+ cells that have departed the PCV, vISVs only, PLs. These show that while there are defects in vessel sprouting and morphogenesis, there is also an overall reduction in the number of cells leaving the PCV (see revised Figure 3B-E).

Finally, we now provide a new Figure 2—figure supplement 2, which examines the early development of the blood vasculature in MZ*yap1* mutants. We find only a mild transient delay in primary angiogenesis in these mutants, providing further support for this phenotype being relatively specific to secondary

2) Further evidence, morpholino versus genetic mutant: Evidence that Vegfc promotes proliferation via Yap1 relies on knockdown of yap1 using a morpholino – and genetic evidence for this important interaction is missing. If mutant lines are not available in the prox1a:KalT4-UAS/UAS:vegfc background, this could be provided using the same transplant approach as in Figure 4 – i.e., upon transplantation of Vegfc-overexpressing cells into control and yap1 mutant embryos and quantification of a resulting increase (or not) of lymphatic cell numbers, as in Figure 4G.

We thank the reviewers for this excellent suggestion. We believe it has significantly improved the manuscript and strengthened the evidence supporting a cell autonomous role for Yap in Vegfc-driven EC proliferation.

The MZ*yap1* mutant was not available on the *Tg(prox1a:KalT4/UAS:vegfc)* background during the revision period, so we instead performed the suggested transplantation experiments. We were able to use blastomere stage transplantations to achieve grafts of single muscle fibres overexpressing Vegfc in host embryos adjacent to the HM in sibling control and MZ*yap1* mutants. In controls, we consistently saw a vast proliferative response of the adjacent venous derived ECs (PLs) in the HM but in MZ*yap1* mutants this was significantly reduced. This experiment was performed several times to achieve high numbers and confidence in this major finding from the study (see new data in Figure 4D-G, n=18 control and n=22 MZ*yap1* successful grafts).

In addition, we have now added an additional experiment in new Figure 5—figure supplement 1. Here we transplanted cells and isolated EC grafts that expressed EGFP-YAP in control or vegfc-OE embryos. Measuring the nuclear concentration of EGFP showed a clear increase in venous or PL grafts in the *vegfc*-OE host embryos. This improved confidence further that both PL proliferation is Yap-dependent and that Vegfc drives increased nuclear Yap.

3) Method and implications of nuclear YAP-EGFP quantifications: Live imaging analyses of EGFP-YAP nuclear accumulation aim to indicate dynamic activation of YAP in lymphatic endothelial cells, but cannot rule out that underlying fluctuations in EGFP-YAP expression may account for observed changes. Transplantation experiments to generate mosaic EGFP-YAP-expressing embryos or mosaic transient expression of EGFP-YAP using Tol2 or likewise would enable dynamic quantification of cytoplasmic to nuclear ratios of single cells and confirm YAP activation dynamics. If these ratios match the nuclear EGFP-YAP to H2B-mCherry ratios for individual cells, then this would provide convincing evidence that further quantification used for Vegfc-stimulation experiments in Figure 4 are appropriate.

We understand the reviewers concerns here and spent considerable time attempting to address this issue. PLs are highly variable in their cytoplasmic distribution and generating grafts of individual PLs for the suggested analyses was extremely challenging. Ultimately, we were not able to use chimeric embryos to generate useful data (data not shown).

We instead revisited the embryos imaged in Figure 1A-F. We used a simple 2D, manual measurement of cytoplasmic EGFP concentration in single Z-sections, we removed the masked nuclear (H2B-mCherry) signal and then generated a cytoplasmic/nuclear concentration index across multiple PLs. Comparison of the nuclear/cytoplasmic ration with our nuclear concentration measurements with Pearson’s correlation test shows a highly statistically significant correlation (see revised Figure 1H). The nature of this correlation can also be appreciated by the readers/reviewers by comparing the raw data presented in revised Figure 1F and Figure 1—figure supplement 1A-B. We believe this does address the questions raised above and justifies our use of nuclear concentration measurements throughout the rest of the paper.

Having said the above, we also think it is important to note the caveat that Yap is regulated on more levels than just nuclear-cytoplasmic shuttling (Hippo cascade dependent). YAP turnover and degradation could also play a role here if Vegfc-signalling impacted it (e.g. perhaps reducing YAP degradation). We think that investigating this is beyond the scope but to note this caveat in the text we have added the following to the now extended Discussion:

“Moreover, in vivo live imaging showed that dynamic changes occur in EGFP-YAP protein concentration in lymphatic progenitor nuclei. […] Importantly, a high concentration of nuclear EGFP-YAP was promoted by a local source of Vegfc, concomitant with increased PL proliferation (Figure 5 and Figure 5—figure supplement 1).”

Along the same inquiry line, the reviewers point out: In Figure 1E' and Figure 4C' and E EGFP-Yap intensity appears to be inhomogeniously distributed over the nucleus delineated by H2B-mCherry. How does that change in z and wouldn't that observation impact on the quantitative evaluation based on a single confocal plane as described for Figure 1D-E.

We now include analyses of this data with 2 independent methods. We use the same single plane analysis as in the original submission (now in Figure 1—figure supplement 1) and a new analysis of average pixel intensity in 3D across the entire nucleus (masked using the H2B-mCherry signal (see Figure 1F). Analysis of correlation between the 2 methods revealed that they both yield highly concordant results and improve our confidence in this approach (new Figure 1—figure supplement 1E).

In Figure 4H and I, identical treatment conditions have been evaluated after continuous imaging (H) or by acquisition of a single stack, why? If phototoxicity is addressed, please comment in the text.

The single stage analysis was performed to rule out that any change (however unlikely) might be due to bleaching or toxicity during long-term time-lapse. While we did not have any major concern about this, we simply thought it best to be thorough. We have included the following explanation in the revised text:

“This was observed using time-lapse videos (Figure 5H, not shown) as well as acquiring images following a single 12-hour treatment to exclude interference from time-lapse bleaching (Figure 5I).”